

# Applications and prospects of phosphoproteomics in renal disease research

XueJia Zheng[1], LingLing Zhou[2], TianTian Xu[2], GuoYing Wang[2], YaLi Peng[2], ChunMei Wen[2], MengYao Wu[2], HuiHui Tao[2,3,4] and Yong Dai[1,2,3,4]

[1] The First Affiliated Hospital of Anhui University of Science and Technology, Huainan, Anhui, China
[2] School of Medicine, Anhui University of Science and Technology, Huainan, China
[3] Anhui Province Engineering Laboratory of Occupational Health and Safety, Huainan, Anhui, China
[4] Key Laboratory of Industrial Dust Deep Reduction and Occupational Health and Safety of Anhui Higher Education Institutes, Huainan, Anhui, China

## ABSTRACT

**Introduction:** Phosphoproteomics, an advanced branch of molecular biology, utilizes specific techniques such as mass spectrometry, affinity chromatography, and bioinformatics analysis to explore protein phosphorylation, shedding light on the cellular mechanisms that drive various biological processes. This field has become instrumental in advancing our understanding of renal diseases, from identifying underlying mechanisms to pinpointing new therapeutic targets.

**Areas covered:** This review will discuss the evolution of phosphoproteomics from its early experimental observations to its current application in renal disease research using liquid chromatography-tandem mass spectrometry (LC-MS/MS). We will explore its role in the identification of disease biomarkers, the elucidation of pathogenic mechanisms, and the development of novel therapeutic strategies. Additionally, the potential of phosphoproteomics in enhancing drug discovery and improving treatment outcomes for renal diseases will be highlighted.

**Expert opinion:** Phosphoproteomics is rapidly transforming renal disease research by offering unprecedented insights into cellular processes. Utilizing techniques such as LC-MS/MS, it enables the identification of novel biomarkers and therapeutic targets, enhancing our understanding of drug mechanisms. This field promises significant advancements in the diagnosis and treatment of renal diseases, shifting towards more personalized and effective therapeutic strategies. As the technology evolves, its integration into clinical practice is pivotal for revolutionizing renal healthcare.

Corresponding authors
HuiHui Tao, 475433093@qq.com
Yong Dai, daiyong22@aust.edu.cn

## ARTICLE HIGHLIGHTS

- Phosphoproteomics is crucial in revealing the complex mechanisms of cellular signal transduction in renal diseases, essential for diagnosis, elucidating pathological processes, and discovering new therapeutic targets.
- The article detailed the sample processing, data collection, and data analysis processes of phosphoproteomics, from which it derived reasonable prospects that can be applied in renal disease research.
- Enhancing technical sensitivity, simplifying protocols, and developing more powerful data analysis tools are necessary to advance future research in phosphoproteomics.

This article is primarily aimed at a diverse audience within the scientific and medical community, including academic researchers and students who are focused on molecular biology and biochemistry. These readers are likely interested in the technical details and experimental methodologies of phosphoproteomics. Clinical researchers and nephrologists are also a significant part of the audience, as the findings have direct implications for developing new therapeutic strategies and diagnostic tools for renal diseases. Additionally, professionals in the pharmaceutical and biotechnology sectors would find the discussion of new biomarkers and therapeutic targets particularly relevant for drug development and innovation in treatment technologies. The comprehensive nature of the article, combining detailed scientific methodologies with clinical applications, makes it a valuable resource for those involved in both the research of kidney diseases and the practical aspects of medical treatment and healthcare policy planning.

## SURVEY/SEARCH METHODOLOGY

In preparing this review article on the application of phosphoproteomics in renal diseases, several databases were selected—PubMed, Web of Science, Scopus, and Google Scholar—to ensure a broad coverage of academic resources. Subsequently, a comprehensive search was conducted across these databases, screened the results to identify relevant studies, and ultimately compiled the review using the most pertinent and high-quality sources from all the selected databases. My search strategy involved carefully chosen keywords related to phosphoproteomics and renal pathology, such as "phosphoproteomics", "renal disease", "kidney", and "phosphorylation", combined with Boolean operators to refine the results. To enhance the precision of my searches, we specifically utilized medical subject headings (MeSH) terms in PubMed. In selecting literature, clear inclusion and exclusion criteria were set. Only studies that employed well-defined experimental procedures, provided detailed data analysis methods, and had systematic and transparent methodologies were included. Studies were excluded if they had methodological ambiguities, small sample sizes (fewer than 10 participants or samples), or were not peer-reviewed. The publication timeframe was restricted the most recent 5 to 10 years to ensure the inclusion of the latest research findings. Additionally, EndNote was used to organize and manage all relevant literature, which not only facilitated effective citation but also ensured systematic progress. Throughout the review writing process, ongoing attention was paid to the latest research,

regularly updating my literature search to ensure that my review was both comprehensive and timely.

## INTRODUCTION

Protein phosphorylation is a crucial post-translational modification process that involves the addition of phosphate groups to specific amino acid residues of proteins (*Gonçalves et al., 2018*). This process, primarily regulated by protein kinases and phosphatases, forms the core of cellular signaling networks and plays a vital role in maintaining normal cellular functions and responding to environmental changes (*Pang et al., 2022*). Phosphorylation not only affects the structure and function of proteins but also influences their cellular localization and stability. It plays a significant role in various biological processes such as cell signal transduction (*Proud, 2019*), gene expression (*Soma et al., 2021*), cell cycle regulation (*Wang et al., 2021*), and is closely associated with the development of diseases in systems like the nervous and urinary systems (Fig. 1).

Phosphoproteomics is a technology integrated with systems biology, dedicated to studying the global landscape of protein phosphorylation modifications (*Riley & Coon, 2016*). By analyzing phosphorylation networks and protein interaction networks, phosphoproteomics offers powerful tools and new perspectives for revealing complex intracellular signaling mechanisms. With the continuous advancements in technology and enhancements in data analysis capabilities, the application of phosphoproteomics is anticipated to become increasingly widespread, playing a more significant role in biomedical research (*Yates, Ruse & Nakorchevsky, 2009*). This trend is particularly evident in the field of mass spectrometry. The evolution of mass spectrometry (*Schaffer et al., 2019*) techniques, including higher sensitivity, broader dynamic range, and faster data acquisition speed, has enhanced the detection and quantitative analysis of phosphorylated peptides, enabling the quantification of more phosphorylated sites and improving site localization accuracy (*Zhang et al., 2013*; *Gibbons et al., 2015*). Over time, improvements in mass spectrometry instruments have enabled the detection and quantification of a wider range of phosphopeptides, including those at lower abundances, which are crucial for studying the phosphorylation states of proteins and their roles in cellular signaling (*Gerritsen & White, 2021*). Moreover, advances in mass spectrometry data analysis software and algorithms have specifically improved the identification of phosphopeptides from mass spectrometry data and enhanced the functional analysis of the resulting datasets (*Gibbons et al., 2015*; *Halder et al., 2021*).

The kidney is a vital excretory organ in the human body, with primary functions (*Venkatakrishna et al., 2023*) that include excretion, regulation of body fluid and electrolyte balance, maintenance of blood pressure stability, promotion of red blood cell production, acid-base balance, and participation in regulating the body's endocrine system, among other physiological functions. Protein phosphorylation plays an essential role in maintaining these functions, including but not limited to regulating the reabsorption and excretion functions of renal tubules, transmitting renal hormone signals, participating in the repair process of renal damage, and regulating metabolic balance in the

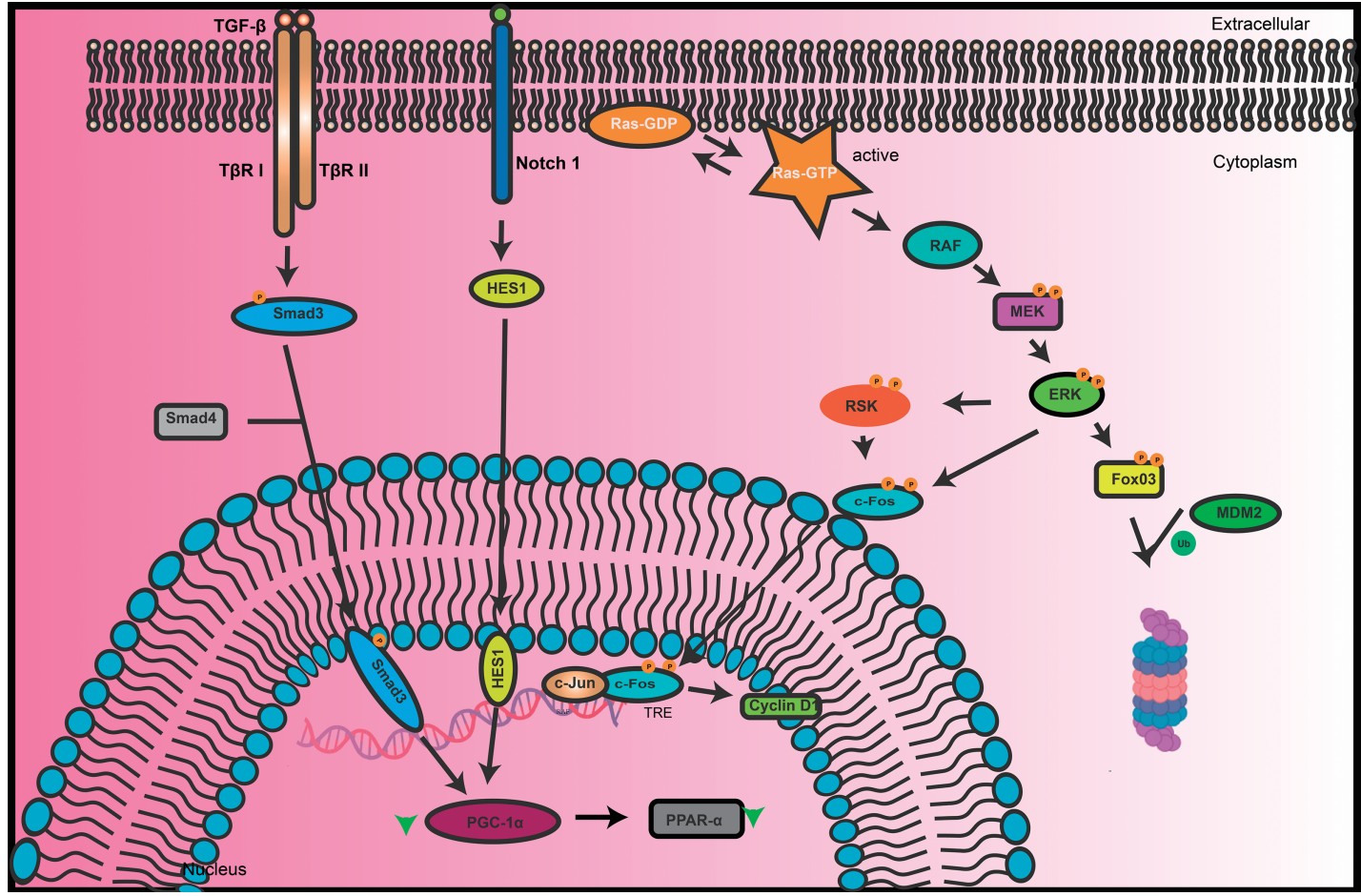

**Figure 1 The key signal transduction processes of TGF-β and Notch signaling pathways in renal cells.** When TGF-β binds to its receptors (TβR I/II), it activates downstream signals through the Smad3/4 complex. Meanwhile, the Notch1 signaling pathway interacts with the Ras-RAF-MEK-ERK pathway through the HES1 transcription factor. Upon activation, Ras-GTP activates RAF, which in turn activates MEK and subsequently ERK. ERK can phosphorylate multiple substrates, including transcription factors such as RSK and c-Fos, which then regulate gene expression in the nucleus. Additionally, ERK activation influences the activity of Fox03, and MDM2, which regulates protein stability. The integrated regulation of these signaling pathways plays a crucial role in maintaining renal phosphate homeostasis, cell proliferation, differentiation, and overall renal function.

body (*Borza et al., 2022*). Therefore, in-depth research into protein phosphorylation holds immense potential for uncovering the mechanisms of kidney disease, identifying new diagnostic markers, assessing treatment outcomes and prognosis, discovering new therapeutic targets, implementing personalized treatment plans, and studying the mechanisms of drug action. Protein phosphorylation is a critical regulatory mechanism in many cellular signaling pathways and is closely related to biological processes such as cell proliferation, differentiation, apoptosis, and metabolism. Studies have shown that in diabetic mouse models, the phosphorylation levels of microtubule-associated protein 4 (MAP4) are elevated, leading to reorganization of the cytoskeleton and changes in the phenotype of podocytes, which results in proteinuria. These findings highlight the central role of the p38/MAPK signaling pathway in diabetic nephropathy and suggest it as a new potential therapeutic target (*Li et al., 2022*).

## PHOSPHOPROTEOMICS

In recent years, phosphoproteomics has undergone remarkable advancements through the progressive integration of innovative computational and experimental techniques, significantly enhancing both the accuracy and efficiency of phosphoproteome analysis.

The foundation for modern phosphoproteomics was laid by *Johnson et al.*'s *(2023)* creation of a comprehensive substrate specificity atlas for the human serine/threonine kinome, which provided an invaluable resource for studying kinase-substrate interactions and signaling networks. Building on this understanding, *Ochoa et al. (2020)* expanded our knowledge by providing crucial insights into the functional landscape of the human phosphoproteome, deepening our understanding of how phosphorylation regulates various biological processes.

A significant breakthrough in experimental workflows came with *Leutert et al.*'s *(2019)* development of the R2-P2 system, which introduced a rapid and automated phosphoproteomics workflow particularly suited for complex signaling studies. This automation was furhr refined by *Bortel et al. (2024)* who systematically optimized the phosphopeptide enrichment process, significantly improving the sensitivity of phosphoproteomic analyses and enabling the detection of low-abundance phosphosites (*Bortel et al., 2024*). Complementing these advances, *Humphrey et al. (2018)* introduced the EasyPhos platform, providing a robust framework for high-throughput and high-sensitivity phosphoproteomic experiments, which opened new possibilities for large-scale studies.

In parallel with these experimental developments, computational approaches saw significant progress. *Luo et al. (2019)* introduced DeepPhos, a deep learning-based tool that revolutionized the prediction of protein phosphorylation sites, leading to substantial improvements in prediction accuracy. This computational advancement was followed by *Lou et al.*'s *(2021)* development of DeepPhospho, which accelerated phosphoproteomics analysis by generating *in silico* spectral libraries, particularly beneficial for data-independent acquisition (DIA)-based workflows. Building on these computational tools, *Bekker-Jensen et al. (2020)* introduced an advanced method for deep phosphoproteome profiling that bypassed the need for spectral libraries altogether, streamlining DIA-based analyses and making them more accessible for large-scale studies.

Recent innovations have focused on integrating multiple approaches and improving data analysis. *Martínez-Val et al.*'s *(2023)* Hybrid-DIA technology represents a significant advancement by combining both targeted and discovery-based approaches, enabling in-depth analysis of phospho-signaling in single-cell systems like spheroids. The field's analytical capabilities were further enhanced by *Yılmaz et al.*'s *(2021)* approach for kinase activity inference using functional networks and *Kuleshov et al.*'s *(2021)* KEA3 tool, which improved kinase enrichment analysis through data integration. Most recently, *Krug et al. (2019)* provided a curated resource for analyzing phosphosite-specific signatures, advancing our ability to link phosphosite modifications with functional outcomes.

## Sample preparation

During the collection and processing of samples, it is imperative to proceed rapidly and under low-temperature conditions to minimize the activity window of phosphatases. Samples that are not processed immediately should be rapidly frozen and stored at −80 °C to ensure effective inhibition of enzymatic activity. These measures not only protect the phosphorylation state of the samples but also ensure the reliability and reproducibility of experimental data, providing a solid foundation for accurate analysis of protein functions.

In successful phosphoproteomics research, sample preparation is a critical consideration (*Paulo & Schweppe, 2021*). Specific requirements must be followed when handling cell and tissue samples, as well as biological fluids such as plasma, cerebrospinal fluid, urine, and saliva (*Moaddel et al., 2021*; *Skoczylas et al., 2023*). In proteomics and molecular biology, inhibiting the activity of endogenous proteases, as well as phosphatases, is a crucial step to ensure the accuracy of the research. The primary aim is to prevent nonspecific changes in the phosphorylation states of proteins within the samples. During experiments, specific phosphatase inhibitors are commonly added to the sample processing or cell lysis buffers (*He et al., 2023*). These inhibitors include sodium fluoride (NaF) (*Schwarz et al., 2023*), sodium pyrophosphate ($Na_4P_2O_7$) (*Zhu, Mehl & Cooley, 2022*), vanadates (*Sun, 2019*), and Calyculin A. Calyculin A is a specific inhibitor, primarily targeting serine/threonine protein phosphatases (*Suganuma et al., 1990*). Additionally, the use of heat to inactivate phosphatases could also be considered as an alternative method.

Due to the low stoichiometry of phosphorylation, meaning there is a relatively low proportion of phosphorylated proteins compared to non-phosphorylated proteins in the proteome, various methods have been developed to enrich these peptides. These include immobilized metal ion affinity chromatography (IMAC) (*Yan, Zhang & Deng, 2014*; *Yang et al., 2017*), metal oxide affinity chromatography (MOAC), and the use of polydopamine-derived mesoporous channels for loading Ti(IV) (*Xu et al., 2021*) ions and immunoprecipitation. However, specificity in the enrichment process has always been a challenge (*Engholm-Keller & Larsen, 2013*). To address the challenge of low specificity in the enrichment of phosphorylated peptides, researchers have effectively reduced nonspecific binding and significantly improved enrichment specificity. This was achieved by adjusting the pH of the loading buffer or using different organic acids, solvents, and additives. Enhancing specificity is crucial, as low specificity can lead to high background noise, which obscures the detection of target peptides and complicates data interpretation (*Carregari, 2022*; *Larsen et al., 2005*).

## Mass spectrometry

Data-dependent acquisition (DDA) and data-independent acquisition (DIA) (*Lou et al., 2021*) are the two primary strategies for mass spectrometry data collection that have dominated the field (*Fernández-Costa et al., 2020*). DDA, a traditional and widely used strategy, involves sequential steps: liquid chromatography separation of phosphopeptides, full-scan mass spectrometry detection, selective fragmentation of the most abundant parent ions, and MS2 scanning (*Ten-Doménech et al., 2020*). Peptides are separated during the liquid chromatography phase, and the MS1 scan detects all parent ions. The system

automatically selects the most abundant ions for fragmentation, and MS2 scans the resulting fragment ions, providing spectra for peptide identification. DDA is favored for its straightforward operation and effectiveness in initial protein identification and phosphorylation site discovery. However, it faces limitations in detecting low-abundance peptides and ensuring data reproducibility, as different parent ions may be selected for fragmentation in each experiment, affecting the consistency and comparability of results.

In contrast, DIA is an innovative approach that does not rely on MS1 scans for precursor ion selection (*Chen et al., 2017*). Instead, DIA fragments all peptide precursor ions within a defined mass range, offering comprehensive coverage (*Meier et al., 2020*). This approach overcomes selection biases inherent in DDA (*Frankenfield et al., 2022*), allowing for more reproducible and complete data acquisition. Although DIA requires advanced data analysis software, it has demonstrated superior performance in terms of coverage and quantitative accuracy, as evidenced by studies such as *Bekker-Jensen et al. (2020)*, which highlighted its effectiveness in deep phosphoproteome profiling without the need for spectral libraries (*Bekker-Jensen et al., 2020*). While DIA has stringent requirements for experimental design, its sample preparation is often simpler than DDA-based techniques like TMT or SILAC, as it does not require labeling for quantification.

Advancements in mass spectrometry techniques have also introduced targeted approaches like multiple reaction monitoring (MRM) (*MacDonald et al., 2018*) and parallel reaction monitoring (PRM) (*Pascual & Kangasjärvi, 2022*), designed to enhance the specificity and sensitivity of detecting targeted analytes. MRM monitors specific transitions from parent ions to daughter ions, providing exceptional sensitivity for quantitative analysis of known targets such as biomarkers. PRM, on the other hand, relies on high-resolution and accurate mass determination to monitor all fragment ions in a single scan, offering more comprehensive data and improved peptide identification accuracy. However, PRM is limited to quantifying a small number of target peptides.

Finally, technological advancements like ion mobility spectrometry-mass spectrometry (IMS-MS) (*Liu et al., 2022*; *Trimpin et al., 2019*) provide additional capabilities in molecular separation and data analysis. By adding a separation dimension based on ion shape and mass, IMS-MS addresses high sample complexity and overlapping components, making it particularly valuable in structural proteomics research.

## Data analysis

Data analysis in phosphoproteomics is a complex process aimed at precisely identifying phosphorylated peptides and their sites from mass spectrometry data (*Luo et al., 2019*). This process includes several steps: data preprocessing, identification of peptides and phosphorylation sites, quantitative analysis, and bioinformatics analysis (*Xiao, Chen & Yang, 2023*). In the data preprocessing stage, the primary task performed by the software is to convert raw mass spectrometry data into a series of discrete peaks representing the mass-to-charge ratio (m/z) of different ions and their corresponding intensities. This process involves removing background noise and correcting data quality to provide accurate foundational data for subsequent analyses. The identification of phosphopeptides

and the localization of phosphorylation sites involve comparing experimental mass spectrometry data with pre-built theoretical mass spectrometry databases. Specifically, this includes comparing observed mass-to-charge ratios with the predicted mass-to-charge ratios for known protein sequences in the database, matching the experimental peptide mass-to-charge ratios against the theoretical peptide mass-to-charge ratios. Additionally, phosphorylation sites are localized to specific amino acids within the peptide sequence, with the accuracy of localization varying based on the probability of correct localization. This method is particularly relevant for spectrum-centric approaches, including those used in database searches of data-dependent acquisition (DDA) data or when employing data-independent acquisition (DIA)-deconvolution algorithms to generate pseudo-DDA spectra. It is crucial to clarify that when spectral libraries are used, direct searching of DIA data against sequence databases does not utilize this method (*Luo et al., 2019*; *Bekker-Jensen et al., 2020*). The correct setting of search parameters such as the selection of protein sequence databases, enzyme digestion rules, types of modifications (fixed and variable), and peptide and MS/MS mass tolerances is essential to ensure the accuracy of identification. Additionally, controlling the false discovery rate (FDR) (*Käll, Storey & Noble, 2008*) is an important means to assess the credibility of identification results. FDR is a statistical method used to estimate the proportion of false positives among the identified items, such as peptides or proteins. It is typically calculated by comparing the number of false positive identifications made under a null hypothesis to the total number of identifications, thus providing a measure of the reliability of the experimental results. Spectral library matching offers an alternative strategy for identification by comparing experimental data with a validated experimental mass spectrometry library. This library is generated from prior validated experiments, where identified spectra are collected and cataloged to create a reference database. This approach enhances the accuracy of identifications as it allows direct comparison to known, verified spectral profiles, reducing the likelihood of false positives. This method is especially suitable for widely studied samples, such as those from commonly researched biological tissues or frequently analyzed compounds, where abundant reference data are available. Bioinformatics analysis further interprets the function and biological significance of phosphorylated proteins, through functional annotation, pathway analysis, and construction of protein interaction networks, revealing the roles of phosphorylation events in biological processes and their positions within regulatory networks.

## APPLICATIONS OF PHOSPHOPROTEOMICS IN RENAL RESEARCH

By comparing the phosphoproteomes of healthy *vs.* diseased kidneys, it also helps understand how environmental and genetic factors affect renal function (Table 1). The key role of phosphoproteomics in renal disease research is closely linked to its ability to delve into the molecular mechanisms of renal disorders. Through phosphoproteomics, researchers can identify signaling pathways that are dysregulated in various renal diseases, aiding in the discovery of biomarkers for early diagnosis and monitoring disease

**Table 1 Comprehensive summary of molecular mechanisms, biomarker identification, and pharmacotherapy advances in phosphoproteomics research of renal disease.**

| Application | | Main finding | Reference |
|---|---|---|---|
| Molecular mechanism studies in renal disease | V2R | V2R activation by dDAVP in rat kidney collecting duct cells modulates protein kinase A-regulated pathways influencing actin dynamics, cell proliferation, and calcium signaling | *MacDonald et al. (2018)* |
| | RIC | RIC does not provide significant protection against ischemia/reperfusion injury. | *Trimpin et al. (2019)* |
| | mTOR | In tuberous sclerosis complex-related polycystic kidney disease, aberrant activation of the mTOR signaling pathway leads to excessive S6 kinase 1 (S6K1) activity, which promotes cyst formation by phosphorylating Afadin and altering its interaction with cell adhesion molecules. | *Salhadar et al. (2021)* |
| | PKC-α | The interaction of Protein Kinase C-α (PKC-α) increases the phosphorylation and activity of F0F1-ATP synthase subunits, enhancing the energy metabolism of renal cells. | *Noda & Sasaki (2021)* |
| | PAK2 | In ccRCC, key phosphorylated proteins such as PAK2, CDK1, and JNK1 that promote tumor proliferation and migration were identified, with high expression of PAK2 associated with poorer survival outcomes. | *Leo et al. (2022)* |
| | COVID-19 | An increase in eosinophilic, tubule-like structures within the tubular lumens of kidneys in mice infected with COVID-19. | *Rinschen et al. (2022)* |
| | mTOR & AMPK | Metabolic reprogramming in kidney damage caused by hypertension involves enhanced lipid breakdown and activation of compensatory pathways in early-stage damage, including changes in mTOR and AMPK signaling in the glomeruli. | *Franzin et al. (2021)* |
| Identification of potential biomarkers | uEVs | Validated phosphorylated aquaporin-2 and glycogen synthase kinase-3β as potential key biomarkers for diabetic nephropathy. | *Hausenloy & Yellon (2016)* |
| | uEVs | Using the EVtrap method, PolyMAC enrichment, and GPF-DIA analysis advanced the isolation and identification of phosphorylated proteins in urinary extracellular vesicles. | *O'Brien et al. (2022)* |
| Research on drugs for renal diseases | SGLT2 inhibitors | SGLT2 inhibitors reduce the exposure to these toxins in the body and need for renal detoxification, thereby decreasing the microbial production of uremic toxins. | *Liu et al. (2023)* |
| | VCM | The close association between VCM-induced nephrotoxicity and the peroxisome and PPAR signaling pathways. | *Rinschen et al. (2019)* |
| | G-1 | G-1 and its affected pathways may have a positive effect on inhibiting the growth and spread of renal cancer cells and overcoming patients' resistance to sunitinib treatment. | *Behrens et al. (2020)* |
| | PFKFB4 | The phosphorylation of PFKFB4 by Nuclear Receptor Coactivator 3 (NCOA3) modulates the activity of the pentose phosphate pathway (PPP) through a regulatory loop involving FBP1. | *Li et al. (2023)* |
| | ESAs | Recombinant human erythropoietin (rHuEPO) and newer erythropoiesis-stimulating agents like peginesatide share key signaling pathways in promoting erythropoiesis. | *Rozanova et al. (2021)* |

progression. Additionally, this technique reveals the molecular basis of drug action and resistance, providing potential therapeutic targets for precision medicine.

## Molecular mechanism studies in renal disease

Vasopressin receptor 2 (V2R) is a G protein-coupled receptor (GPCR) that, when activated, can activate adenylyl cyclase type 6, increasing the levels of cyclic AMP (cAMP) in the collecting duct cells of the renal tubules (*Salhadar et al., 2021*). In this process, cAMP plays a crucial role in regulating fluid balance in renal collecting duct cells by regulating the translocation of aquaporin-2 (AQP2) (*Noda & Sasaki, 2021*). *Leo et al. (2022)* and other
researchers utilized quantitative phosphoproteomics technology to track phosphorylation changes induced by the V2R selective agonist dDAVP in rat renal collecting duct cells. They employed a Bayesian approach to integrate dynamic phosphorylation data with multiple previous omics datasets, revealing three main protein kinase modules regulated by protein kinase A (PKA). These modules are involved in controlling actin skeleton dynamics through Rho/Rac/Cdc42-dependent protein kinase pathways, cell proliferation through mitogen-activated protein kinase and cell cycle-dependent protein kinase pathways, and calcium/calcineurin-dependent signaling processes. This study provides significant insights into understanding the signal transduction mechanism of V2R in renal collecting duct cells and reveals a new set of protein kinases associated with the V2R response. Moreover, it proposes a new approach to study complex signal transduction networks.

In the biology of glomeruli, the phosphorylation of signaling proteins, such as cytoskeletal regulators and glomerular podocyte slit membrane proteins, is strictly controlled by signal transduction networks. Rinschen et al. (2022) performed phosphoproteomics analysis on bovine and rat glomeruli, enabling cross-species comparisons. They discovered phosphorylation sites with likely significant biological implications, such as the tyrosine phosphorylation of the cytoskeletal regulator Synaptopodin and the slit membrane protein Neph-1 (Kirrel) (Rinschen et al., 2022). Remote ischemic preconditioning (RIC) has been proposed as a therapeutic intervention to mitigate ischemia/reperfusion injury (IRI) during organ transplantation (Franzin et al., 2021). O'Brien et al. (2022) performed RIC on porcine renal transplant recipients. In a kidney transplant model, the phosphorylation levels showed no significant changes between kidneys that underwent ischemic preconditioning and the untreated control group. A global pool comparison of all eight ischemic preconditioning (RIC) samples and eight non-ischemic preconditioning (non-RIC) control samples identified 3,524 phosphorylation sites across 3,626 proteins. However, no significant differences in phosphorylated protein expression were found between the RIC and non-RIC groups, indicating that RIC had minimal steady-state effects on kidney tissue at this specific time point (Hausenloy & Yellon, 2016; O'Brien et al., 2022).

In the study of tuberous sclerosis complex (TSC), the aberrant activation of the mammalian target of rapamycin (mTOR) signaling pathway is considered a key factor in the pathogenesis of polycystic renal disease (O'Brien et al., 2022). This pathological condition is characterized by excessive cell proliferation and cyst formation, closely associated with the loss of oriented cell division (OCD) (Pema et al., 2016) functionality. The discoveries made by Bonucci et al. (2020) have further unveiled the molecular mechanisms behind this process, highlighting that the activation of S6 kinase 1 (S6K1) fundamentally distorts OCD. In mouse models with Tsc1 gene mutations, the absence of S6K1 was able to restore the normal function of OCD, yet it failed to inhibit excessive cell proliferation, leading to non-cystic renal growth. Through phosphoproteomics analysis based on mass spectrometry of S6K1 substrates, researchers identified Afadin, a cellular adhesion component capable of coupling intercellular adhesion and cortical cues with spindle orientation (Bonucci et al., 2020). S6K1 directly phosphorylates Afadin, causing

abnormal modifications with E-cadherin and α-catenin. The excessive activity of S6K1 not only alters the positioning of the centrosome in mitotic cells but also disrupts the process of oriented cell division, thereby promoting the formation of renal cysts under the condition of overactive mTOR signaling pathway. This study underscores the central role of the mTOR (*Pema et al., 2016*) signaling pathway in TSC-induced polycystic renal alterations and reveals the critical roles of S6K1 and its substrate Afadin in regulating oriented cell division and maintaining the structural stability of renal tissue.

*Nowak & Bakajsova (2015)* employed phosphoproteomics analysis in conjunction with a suite of advanced biochemical and molecular biology techniques, including renal tubular cell culture, adenovirus-mediated gene transfection, immunoprecipitation, and ATP synthase activity assays, to thoroughly investigate the interaction between Protein Kinase C-α (PKC-α) and F0F1-ATP synthase subunits and its regulatory effect on ATP synthase function. They discovered that PKC-α, by interacting with, could enhance the phosphorylation levels of F0F1-ATP synthase subunits, thereby promoting an increase in ATP synthase activity. This finding highlights the crucial role of PKC-α in maintaining the energy metabolic balance of renal cells and offers potential new strategies for the treatment of renal diseases (*Nowak & Bakajsova, 2015*). *Senturk et al. (2022)* conducted a comprehensive quantitative phosphoproteomics analysis of tumor tissues and their adjacent normal tissues in clear cell renal cell carcinoma (ccRCC), uncovering significantly upregulated phosphorylated proteins closely associated with tumor proliferation and migration behaviors. Notably, p21-activated kinase 2 (PAK2), cyclin-dependent kinase 1 (CDK1), and c-Jun N-terminal kinase 1 (JNK1) played pivotal roles in this process. Moreover, the high expression of PAK2 was linked to poorer survival outcomes in ccRCC patients, providing a theoretical basis for new therapeutic targets (*Senturk et al., 2022*).

*Liu et al. (2023)* utilized phosphoproteomics studies to analyze renal function in mice infected with COVID-19, identifying a pathological phenomenon characterized by an increase in eosinophilic, tubule-like structures within the tubular lumens of the infected renals. *Rinschen et al. (2019)*, through the integrated omics approach combining phosphoproteomics, metabolomics, and proteomics, conducted an in-depth investigation into the effects of hypertension on the renals, particularly on glomerulosclerosis, uncovering the mechanisms of metabolic reprogramming behind hypertension-induced renal damage. The study revealed that early-stage renal damage due to hypertension involves intensified lipid breakdown and the activation of compensatory metabolic pathways to counteract the metabolic stress induced by hypertension, including decreases in ATP and NADH levels and an increase in oxidized lipids. Notably, in the glomeruli, the activation of mTOR and AMPK signaling pathways regulated key molecules in podocytes, such as cytoskeletal components and GTP-binding proteins. This regulation reflects specific changes in metabolic signaling, including alterations in glucose uptake and lipid metabolism (*Rinschen et al., 2019*).

## Identification of potential biomarkers for renal diseases

*Li et al. (2023)* employed a quantitative phosphoproteomics method to study phosphorylated proteins within urinary extracellular vesicles (uEVs), leading to the

detection of 233 phosphorylated peptides (*Behrens et al., 2020*). The experiment identified 233 phosphorylated peptides in uEVs, 47 of which demonstrated significant changes in phosphorylation between patients with diabetic nephropathy (DN) and those with diabetes but without nephropathy. To validate these findings, they further analyzed phosphorylated aquaporin-2 (p-AQP2[S256]) and phosphorylated glycogen synthase kinase-3β (p-GSK3β[Y216]). Through Phos-tag Western blotting and immunohistochemical staining of renal sections, the presence of these two phosphorylated proteins in uEVs was successfully confirmed. These discoveries suggest that changes in phosphorylated proteins within uEVs can reflect corresponding alterations in the renal and have the potential to serve as candidate biomarkers for DN, offering a highly feasible tool for the diagnosis of renal diseases (*Rozanova et al., 2021*).

*Hadisurya et al. (2023)* made significant advancements in the isolation and identification of phosphorylated proteins within urinary extracellular vesicles (uEVs). Utilizing the extracellular vesicle trapping (EVTrap) method, combined with polyvalent metal affinity chromatography (PolyMAC) enrichment and gas-phase fractionation data-independent acquisition (GPF-DIA) analysis, they successfully isolated and identified thousands of unique phosphorylation sites. This research achievement has the potential to enable urologists to better determine optimal treatment methods and provides more rational treatment plans for newly diagnosed patients with low-grade clear cell renal cell carcinoma. By actively monitoring rather than prematurely undertaking surgeries that could lead to overtreatment and excessive costs, patient burden can be alleviated, and more accurate medical management can be offered (*Hadisurya et al., 2023*).

## Research on drugs for renal diseases

Sodium-glucose cotransporter 2 inhibitors (SGLT2i) are effective medications for managing hyperglycemia in patients with type 2 diabetes and also offer renal protection against failure (*Brito et al., 2020*; *Heerspink et al., 2020*; *Voors et al., 2022*). However, their mechanism of action requires further exploration. In a study conducted by *Billing et al. (2023)* non-diabetic mice without early-stage hyperglycemia were treated with an SGLT2 inhibitor for 1 week, followed by comprehensive proteomics, phosphoproteomics, and metabolomics analyses. The results indicated that SGLT2 inhibitors significantly protect the renals of non-diabetic mice, evidenced by reduced glycoxicity in the proximal tubules and downregulation of various uptake transport mechanisms, including the intake of sodium, glucose, uric acid, purine bases, and amino acids. SGLT2 inhibitors reduce the exposure to these toxins in the body and the need for renal detoxification, thereby lowering the microbial production of uremic toxins and providing an important metabolic foundation for renal protection. This study provides significant scientific evidence for a deeper understanding of the pharmacological actions of SGLT2 inhibitors (*Billing et al., 2023*).

*Yang et al. (2023)* conducted a comprehensive study into the nephrotoxicity induced by vancomycin (VCM). Through phosphoproteomics analysis of mouse kidney tissues, they identified 3,025 phosphopeptides showing changes in phosphorylation, indicating shifts in protein activity and signaling pathways. Gene ontology enrichment analysis revealed that

oxidoreductase activity and peroxisomes play pivotal roles in VCM-induced nephrotoxicity. Moreover, KEGG pathway analysis showed an enrichment of the peroxisome pathway and PPAR signaling pathway. Further parallel reaction monitoring (PRM) analysis indicated that VCM significantly downregulated the phosphorylation levels of proteins associated with fatty acid β-oxidation in the PPAR signaling pathway. This study highlights the close association between VCM-induced nephrotoxicity and the peroxisome and PPAR signaling pathways, providing crucial theoretical support for a deeper understanding of the regulatory mechanisms of nephrotoxicity and potential therapeutic strategies (*Yang et al., 2023*).

For patients with metastatic renal cell carcinoma (RCC), first-line targeted treatments such as sunitinib often produce resistance, impacting the effectiveness of the therapy. *Chen et al. (2021)* treated parental and sunitinib-resistant 786-O cells with G protein-coupled estrogen receptor 1 (GPER1) agonist G-1 and conducted a quantitative phosphoproteomics study. The results indicate that G-1 and its affected pathways significantly inhibit the growth and spread of renal cancer cells and enhance patients' responsiveness to sunitinib treatment, providing new insights and strategies for the treatment of renal cancer (*Chen et al., 2021*). *Feng et al. (2021)* discovered that 6-phosphofructo-2-kinase/fructose-2,6-bisphosphatase 4 (PFKFB4) is closely associated with the activity of the Pentose Phosphate Pathway (PPP). Through comprehensive examination and detailed evaluation of its phosphorylation process, it was found that this association is mediated by the phosphorylation process facilitated by Nuclear Receptor Coactivator 3 (NCOA3). The interaction between NCOA3 and FBP1 counteracts the excessive flow of the PPP, creating a regulatory loop. This discovery provides potential targets for the development of new drugs to combat resistance to sunitinib (*Feng et al., 2021*).

In the field of erythropoiesis, prior to the advent of recombinant human erythropoietin (rHuEPO), patients required frequent red blood cell transfusions for treatment (*Green et al., 2012*). With the emergence of rHuEPO, patients gained a more convenient treatment option. However, due to the need for frequent injections of rHuEPO, researchers began to develop erythropoiesis-stimulating agents (ESAs) with extended half-lives through post-translational modifications, thereby reducing the frequency of injections. Green and others conducted a global phosphoproteomics and transcriptomics study to comprehensively reveal the signaling pathways of rHuEPO and new ESAs (such as peginesatide). Despite structural differences, the study found that they exhibit notable similarities in promoting erythropoiesis, indicating that they share some major signaling pathways. These findings are significantly important for understanding and treating chronic renal disease-related anemia and formulating appropriate treatment strategies.

## SUMMARY AND OUTLOOK

Despite the tremendous potential phosphoproteomics has shown in renal disease research, its practical application faces two major challenges: technical complexity and operational efficiency. The primary challenge stems from the low abundance of phosphorylated peptides, requiring highly sensitive methods or increased input material (*Peng et al., 2018*),

while the complexity of phosphorylation site dynamics demands advanced data analysis methods and algorithms (*Valverde et al., 2023*). To address these challenges, future research needs to focus on enhancing technical sensitivity and streamlining operational procedures.

### Funding
The authors received no funding for this work.

### Competing Interests
The authors declare that they have no competing interests.

### Author Contributions
- XueJia Zheng conceived and designed the experiments, performed the experiments, analyzed the data, prepared figures and/or tables, authored or reviewed drafts of the article, and approved the final draft.
- LingLing Zhou conceived and designed the experiments, analyzed the data, authored or reviewed drafts of the article, and approved the final draft.
- TianTian Xu conceived and designed the experiments, authored or reviewed drafts of the article, and approved the final draft.
- GuoYing Wang conceived and designed the experiments, analyzed the data, prepared figures and/or tables, and approved the final draft.
- YaLi Peng conceived and designed the experiments, analyzed the data, prepared figures and/or tables, and approved the final draft.
- ChunMei Wen conceived and designed the experiments, prepared figures and/or tables, and approved the final draft.
- MengYao Wu conceived and designed the experiments, performed the experiments, prepared figures and/or tables, and approved the final draft.
- HuiHui Tao conceived and designed the experiments, performed the experiments, authored or reviewed drafts of the article, and approved the final draft.
- Yong Dai conceived and designed the experiments, authored or reviewed drafts of the article, and approved the final draft.

### Data Availability
This is a literature review.

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
