# Peer review of "Applications and prospects of phosphoproteomics in renal disease research"

_PeerJ, doi:10.7717/peerj.18950_

## Round 0.1 · original submission · Major Revisions

Dear Dr. Yong,

If you feel that you can appropriately revise your paper in response to the reviewers' comments, please submit a revised manuscript. Please also include a response to each of the reviewers' comments, explaining how you have revised the manuscript in response to the reviewers' comments.

Yours,

Yoshi

Prof. Yoshinori Marunaka, M.D., Ph.D.
Academic Editor
PeerJ Life & Environment

Reviewer 1 ·

Basic reporting

The review article entitled “Applications and prospects of phosphoproteomics in renal disease research” provides an overview of mass-spectrometry based proteomics, with focus on phosphoproteomics, and its application in the field of renal disease. Although the review is comprehensive, overall it seems that many relevant papers in the field are missing in this literature review, such as:

- references to the latest works in phosphoproteomics using DIA:
o Luo F, Wang M, Liu Y, Zhao XM, Li A. DeepPhos: prediction of protein phosphorylation sites with deep learning. Bioinformatics. 2019 Aug 15;35(16):2766-2773.
o Lou, R., Liu, W., Li, R. et al. DeepPhospho accelerates DIA phosphoproteome profiling through in silico library generation. Nat Commun 12, 6685 (2021).
o Bekker-Jensen DB, Bernhardt OM, Hogrebe A, Martinez-Val A, Verbeke L, Gandhi T, Kelstrup CD, Reiter L, Olsen JV. Rapid and site-specific deep phosphoproteome profiling by data-independent acquisition without the need for spectral libraries. Nat Commun. 2020 Feb 7;11(1):787).
- references to publications regarding workflows or technical implementations of phosphoproteomics in the clinical context:
o Bortel P, Piga I, Koenig C, Gerner C, Martinez-Val A, Olsen JV. Systematic Optimization of Automated Phosphopeptide Enrichment for High-Sensitivity Phosphoproteomics. Mol Cell Proteomics. 2024 May;23(5):100754.
o Leutert M, Rodríguez-Mias RA, Fukuda NK, Villén J. R2-P2 rapid-robotic phosphoproteomics enables multidimensional cell signaling studies. Mol Syst Biol. 2019 Dec;15(12):e9021.
o Humphrey SJ, Karayel O, James DE, Mann M. High-throughput and high-sensitivity phosphoproteomics with the EasyPhos platform. Nat Protoc. 2018 Sep;13(9):1897-1916.
o Martínez-Val A, Fort K, Koenig C, Van der Hoeven L, Franciosa G, Moehring T, Ishihama Y, Chen YJ, Makarov A, Xuan Y, Olsen JV. Hybrid-DIA: intelligent data acquisition integrates targeted and discovery proteomics to analyze phospho-signaling in single spheroids. Nat Commun. 2023 Jun 16;14(1):3599.
- Information about data analysis of phosphoproteomics datasets, data bases of kinases substrates, tools for phosphorylated pathway annotation and kinase activity inference:
o Johnson, J.L., Yaron, T.M., Huntsman, E.M. et al. An atlas of substrate specificities for the human serine/threonine kinome. Nature 613, 759–766 (2023).
o Ochoa, D., Jarnuczak, A.F., Viéitez, C. et al. The functional landscape of the human phosphoproteome. Nat Biotechnol 38, 365–373 (2020).
o Yılmaz, S., Ayati, M., Schlatzer, D. et al. Robust inference of kinase activity using functional networks. Nat Commun 12, 1177 (2021).
o Kuleshov MV, Xie Z, London ABK, Yang J, Evangelista JE, Lachmann A, Shu I, Torre D, Ma'ayan A. KEA3: improved kinase enrichment analysis via data integration. Nucleic Acids Res. 2021 Jul 2;49(W1):W304-W316.
o Krug K, Mertins P, Zhang B, Hornbeck P, Raju R, Ahmad R, Szucs M, Mundt F, Forestier D, Jane-Valbuena J, Keshishian H, Gillette MA, Tamayo P, Mesirov JP, Jaffe JD, Carr SA, Mani DR. A Curated Resource for Phosphosite-specific Signature Analysis. Mol Cell Proteomics. 2019 Mar;18(3):576-593.
Inclusion of such references will make the review more useful for scientists both in proteomics and renal research.

Finally, in Sections 3.1 to 3.3, I found that the link between the examples of phosphoproteomics applications were not well related to the context of renal disease, since it was not explained what was the reason in those examples to use phosphoproteomics as a technique.

Overall, the English used throughout the text is clear, professional, and unambiguous, and the article structure is correct. However, the literature references can be improved as stated above. This review could be of potential interest once the improvements in literature references are included in a revised version of the manuscript. Finally, the introduction clearly states the importance of studying phosphoproteomics, but it fails to clearly state its role in renal disease. This could be improved by including specific references to phosphoproteomics applications in renal disease or potential research questions in this field that could be addressed using phosphoproteomics.

Experimental design

The review is well organized and references are adequately cited. However, as stated before, some relevant references in the field are missing and literature review needs to be improved before publishing this manuscript in the current form

Validity of the findings

No comment.

Additional comments

Moreover, I found some punctual statements throughout the text that will require rephrasing or further explanations:
- Line 105 – references for imaging in the phosphoproteomic context are missing to validate the statement.
- Line 115 – missing link or basis for the biological interest of studying phosphoproteomics in renal disease.
- Lines 143 to 145 – this sentence needs rephrasing, it is said that calyculin A is a specific inhibitor, but not to which kind pf protein phosphatases. Also, a reference in missing for this inhibitor.
- Line 144 – what does the “i” stands for in NaPPi?
- Line 159 – some references need to be added of works that evaluate the effect of different experimental parameters (buffer pH, use of competitive non-phosphopeptide binders).
- Line 190 – consider changing the verb “reconstruct” by “deconvolute”
- Line 201 – some references on works using DIA for biomedical research needs to be added here, as well as work related to technological development involved in biomedical applications using DIA-based proteomics.
- Line 203 to 206 – consider rephrasing this section, since it mixes acquisition methods (PRM, SRM, Real-Time Search) with technological implementations (Ion Mobility).
- Line 231 to 233 – this sentence only applies to spectrum centric strategies (either those used in database search of DDA data, or when using DIA-deconvolution algorithms to obtain pseudo-DDA spectra). It needs to be clarified that DIA data is not searched using this strategy when spectral libraries.
- Line 348 – GPF acronym is not defined previously.

Reviewer 2 ·

Basic reporting

1. The review is understandable however, the review would greatly benefit from a more precise vocabulary for more efficient communication. Many generic terms have been used and should be avoided. More details are given below.
2. The litterature references should be improved in the proteomics/phosphoproteomics part. Field specific reference papers should be added instead or reviews of application papers which are cited here. References can be found in the bibliography of some pioneers in the field such as Matthias Mann, Jesper Olsen, Albert Heck, Joshua Coon, Martin Larsen among others. Additionally, the authors should check their reference numbering, it seem that some numbers in the text do not match the right reference in the references.
3. The structure of the article is straightforward and efficient with first a description of the phosphoproteomics workflow, followed by a state of the art of literature using phosphoproteomics for renal research. Table 1 is a good ressource for summing up the bibliography. However figure 1 and 2 do not provide a clear overview of the phosphoproteomics workflow / quantification strategies. More details about the figures are given below.
4. The aim of the review is highly relevant for publishing and making this knowledge available. This review aims at bridging the gap between proteomics scientists and clinitians/ cellular biologist with interest in proteomics collaboration. It describes both the technical aspects linked to proteomics/phosphoproteomics studies and their application for renal research with a review of how proteomics increased the knowledge in renal research. This paper could be beneficial for proteomics scientists investigating into what is known in renal research and clinician curious about phosphoproteomics and how it could contribute to their research.
5. Phosphoproteomics is often reviewed, here making phosphoproteomics knowledge available to clinicians makes the review relevant for publication.
6. Yes.

Experimental design

1. Yes.
2. Yes.
3. The methodology is detailed, however, the authors should precise what they mean by "rigorous methodologies", "reliable data", "unclear methodologies", "very small samples sizes" (l.73/74).
4. l.67 - "I initially selected several databases", can the authors specify what happened afterwards? Was one database selected or does the review originate from all databases?
5. Citations on the phosphoproteomics part should be improved.
6. Yes.

Validity of the findings

1.
2. Yes.
3. l. 24- the development of MS-based proteomics hasn´t been fully developped in the review. A few more sentence on the evolution of the field, which has tremendously evolved in the past 10 years, are needed to cover this area.
4. The conclusion should be improved as to clearly state the challenges inherent to phosphoproteomics and how likely it could be applied in the clinics.
Comments on the summary and outlook:
This paragraph would benefit from a summary sentence about the potential of phosphoproteomics for renal research and how it already contributed to the field before being more critical.
l.407 – Phosphoproteomics requires “highly sensitive methods” or the use of higher input amounts of material.
l.409 – What do the authors refer to with “data analysis techniques and algorithms”? The term techniques is unclear.
l.409-411 – This statement is very true and relevant here. However, it would gain in impact by specifying what would a gain in sensitivity provide, why the operational procedures should be simplified and what would be the point of more powerful data analysis tools? Obtaining more depth, a better quantification accuracy, a better localization of the phosphosites?
l.421 – What do the authors call “in-depth studies”?
l.422 – What “technological development” are the authors referring to?
l.432 – “its integration into clinical practices” – This statement could be developed either here or in the discussion part. Would the authors apply phosphoproteomics in the clinic? Do they think it is possible? Or should phosphoproteomics be limited to the discovery part with diagnostics performed with other less expensive and more robust techniques? Has a MS-based phosphoproteomics workflow already been implemented in the clinic?

Additional comments

l.20 – “cutting edge techniques”. The techniques should be explicitly named.
l.99 – “processing speed” should be replaced by acquisition speed or scanning speed or sequencing speed.
l.100 – “more precise and efficient” – those terms are very generic and should be avoided, especially efficient. Do the authors refer to the quantification of more phosphorylated peptides, a better site localization or something else? Additionally, references should be added there to support the increased precision and efficiency.
l.100 – “Over time, improvements in mass spectrometry instruments have enabled the detection of lower abundance phosphorylated peptides, crucial for studying the phosphorylation states of proteins and their roles in cellular signaling.” Would be improved by by writing “enabled the detection and quantification of a wider range of phosphopeptides, including lower abundant ones, crucial for….”.
l. 104 – “have greatly enhanced the accuracy and depth of data interpretation”. This sentence is very unspecific and should be more precise. Additionally, it would greatly benefit from research papers supporting that claim.
l.104 – “data interpretation” is unclear. Do the authors refer to the identification of phosphopeptides from the MS data or the functional analysis of the resulting data?
l.104 – “Additionally, emerging mass spectrometry imaging techniques allow scientists to visually observe the distribution of phosphorylated proteins at the cellular and even tissue level, providing a powerful tool for studying the spatial heterogeneity of phosphorylated proteins.” Please cite the research paper describing this result.
l.125 – “using mass spectrometry (MS) and liquid chromatography- tandem mass spectrometry (LC-MS/MS)”. Please remove the “mass spectrometry (MS) and” from the sentence. Also, another reference besides a review could be added, there has been many papers carrying out large scale quantitative phosphoproteomics.
l.129 – “Experimental operations involve sample preparation[17], enrichment of phosphorylated peptides[18], chromatographic separation of peptides, and the use of tandem mass spectrometry techniques[19].” The sentence could be technically more precise: “Experimental operations involve sample preparation[17], including a specific enrichment of phosphorylated peptides[18], liquid chromatographic separation of the enriched phosphopeptides, and the use of tandem mass spectrometry techniques[19]for phosphopeptide sequencing.”. References should be added and could include for instance pioneer papers in the field of phosphoproteomics:
Pinkse MW, Uitto PM, Hilhorst MJ, Ooms B, Heck AJ. Selective isolation at the femtomole level of phosphopeptides from proteolytic digests using 2D-NanoLC-ESI-MS/MS and titanium oxide precolumns. Anal Chem. 2004 Jul 15;76(14):3935-43. doi: 10.1021/ac0498617. PMID: 15253627.
Larsen MR, Thingholm TE, Jensen ON, Roepstorff P, Jørgensen TJ. Highly selective enrichment of phosphorylated peptides from peptide mixtures using titanium dioxide microcolumns. Mol Cell Proteomics. 2005 Jul;4(7):873-86. doi: 10.1074/mcp.T500007-MCP200. Epub 2005 Apr 27. PMID: 15858219.

l.135 – Figure 1 is unclear. The legend should be developed to concisely explain the phosphoproteomics workflow. Could the authors explain why large scale samples should be analysed in DDA while small scale samples should be analysed in DIA? “Phosphopeptide processing & amplification” is unclear. Do the authors mean phosphopeptide enrichment? What do the authors mean by “pre-fractionation”? Do they refer to off-line fractionation? In that case it would be worth mentioning fractionation, what it is and how it is performed. Are the authors describing a label-free or label-based DDA workflow?

l.145 – It is important to highlight the need for phosphatase inhibitors as performed by the authors. Maybe the use of heat to inactivate the phosphatase could also be mentioned.

The first and second paragraphs of the 2.1 Sample preparation could be inverted to follow the logical flow of the sample preparation.
l.151 – Low adhesion consumables would be recommended, however, it is to my opinion not a necessary step. Could the authors justify this necessity?
l.153 – The first sentence of the paragraph should be rephrased. Phosphorylation is present at a low stoichiometry meaning there is a low proportion of phosphorylated proteins compared to non phosphorylated proteins.
l.157- Why has it been a challenge. Please add a reference.
l.160 – What would happened with low specificity in the enrichment? Citations should be added for the use of additives including: Larsen, Martin R., Tine E. Thingholm, Ole N. Jensen, Peter Roepstorff, and Thomas J. D. Jørgensen. 2005. “Highly Selective Enrichment of Phosphorylated Peptides from Peptide Mixtures Using Titanium Dioxide Microcolumns.” Molecular & Cellular Proteomics: MCP 4 (7): 873–86.
l.164 – The describtions of both DDA and DIA are unclear and should be improved:
l.166 – “samples” should be replaced by phosphopeptides.
l.167 – “secondary mass spectrometry scanning” is a very uncommon term for MS2.
l.167 – I do not understand the relevance of reference 33 for this sentence.
l.168 – The MS acquisition do not “follow” the LC separation.
l.169 – “parent ions” is not explained.
l.173 – What does “in depth analysis” means?
l.174 – Could the authors please clarify how DDA is easier to “operate” compared to DIA and how it would be the method of choice for “discovery of phosphorylation sites and protein identifications”? It has though been demonstrated that DIA outperforms DDA : Bekker-Jensen, Dorte B., Oliver M. Bernhardt, Alexander Hogrebe, Ana Martinez-Val, Lynn Verbeke, Tejas Gandhi, Christian D. Kelstrup, Lukas Reiter, and Jesper V. Olsen. 2020. “Rapid and Site-Specific Deep Phosphoproteome Profiling by Data-Independent Acquisition without the Need for Spectral Libraries.” Nature Communications 11 (1): 787.
l.178 – Key words such as missing values should be added.
l.176 – To understand the limitation of DDA for the quantification of low-abundant phosphopeptides, the selection of the parent ion should be explained.
l.191-194 – The sentence requires a reference.
l.194 – “Nearly all peptides in a sample” is a unspecific big claim that should be removed or supported by a reference.
l.196 – Even though DIA generates more complex spectra compared to DDA, and thus requires different algorithm to perform peptide identification, the challenge lies on the side of the software developers and not end-users. User friendly software have been developed, making this sentence untrue.
l.199 – Could the authors explain “the high demands of DIA on experimental design”?
As the review is targeted to clinicians, a short description of MS and peptide sequencing would be beneficial for understanding the MS paragraph.

l.202 – Targeted approaches such as MRM, SRM, PRM or IMS are not “new methods” and have been developed for a very long time.
l.206 – RTS are relying on MS3-based methods, which also uses chemical labelling. It is either outside the scope of this review or should be explained.
l.212 – PRM can monitor “all fragment ions” in a single scan/spectra.
l.211 – “Providing more comprehensive information” compared to what? SRM methods? DIA methods? Additionally, it should be mentioned that PRM offers a precise quantification, but over a limited number of target peptides only.
The paragraphs about RTS and IMS are not relevant for the understanding of phosphoproteomics. However, it would be interesting to discuss the difference between discovery using DDA or DIA and diagnostic for which a PRM method would be better suited. If the authors want to discuss new methods they could cite the following research paper allowing for both discovery and targeted analysis:
Martínez-Val, Ana, Kyle Fort, Claire Koenig, Leander Van der Hoeven, Giulia Franciosa, Thomas Moehring, Yasushi Ishihama, et al. 2023. “Hybrid-DIA: Intelligent Data Acquisition Integrates Targeted and Discovery Proteomics to Analyze Phospho-Signaling in Single Spheroids.” Nature Communications 14 (1): 3599.

l.226 – 229- If the authors want to mention data preprocessing, they should specify that those steps are performed by the software.
l.230 – “sequence database” should be replaced by “protein sequence database”.
l.230 – 233 – The explanation of the phopshopeptide identification should be rephrased as it is unclear. It should be explicitely mentioned which experimental metric is matched against which theoretical metric. Additionally, a sentence about how phosphosites are localized, or at least about the need to localize the modification to a specific amino acid, and with different levels of confidence based on the localization probability should be added.
l.236 – The authors are mentioning the FDR, they should then explain what it is and briefly how it is calculated. Additionally, the original paper describing the FDR calculation could be cited: Käll, Lukas, John D. Storey, and William Stafford Noble. 2008. “Non-Parametric Estimation of Posterior Error Probabilities Associated with Peptides Identified by Tandem Mass Spectrometry.” Bioinformatics 24 (16): i42–48.
l.238 – “comparing experimental data with a validated experimental mass spectrometry library” the authors should explain how this library is generated, and how it enhance the “accuracy” of identifications. Which samples do the authors refer to with “widely studied samples”?
l.240 – If the authors do not explain labelling strategies, they should not mention it here.
l.241 – it is actually the quantitative precision of the measurement which is key for ensuring the accuracy of the statistical analysis and not the other way around, which in turn guarantee reliable interpretation of the results.

Applications of phosphoproteomics in renal research:

Altogether this paragraph is of interest to review the current status of phosphoproteomics for renal research. Table 1 is a great tool for summarizing this part.
l.248 – 250 – Has already been said in the introduction.
l.256 – “by comparing the phosphoproteomes of healthy versus diseased renals” is very important and do not only refer to the understanding of the impact of environmental and genetic factors, it should be moved up in this paragraph as it is the basis of any experimental design. An appropriate control has to be used as relative quantification is performed. Therefore, it is crucial for all the applications named above in the paragraph: cell signaling, biomarker discovery, drug action and resistance,…
l.278 – “potentially high biological significance” should be rephrased with precise terms.
l.280-286 – Here, the authors are summing up results from transcriptome and proteome analysis. They should either remove this part, discuss the phosphoproteomics results, or link the proteomics/transcriptomics results with phosphoproteomics results. Additionally reference 50 is a review and the original research paper should be cited as well.
l.310 – “through interaction” should be rephrased.
l.333 – Examples of “specific changes in metabolic signaling” should be named.
l.337 – the authors mention l.335 “ultra-high sensitivity quantitative phosphoproteomics method” but it leads to the detection of “233 phosphorylated peptides”, which seem extremely low. The authors could remove the statement l.335 or give more information about how much sample was used for the analysis and the method used to justify the statement l.335.
l.345 – I do not understand the choice of reference 59 here.
l.347 – The abbreviation uEV has been introduced in the first paragraph and should be used here too. Could “EVTrap”, “PolyMAC” and “GPF-DIA” be explained? Or at least the abbreviation should be explicated and citations explaining the techniques should be added.
l.349 – “This research achievement enables urologists to better determine optimal treatment” – Does it mean that it is used at the moment in the clinic?
l.359 – “in-depth proteomics” – What do the authors mean by that?
l.360-367 – The paragraph would be more impactful for this review if the authors would specify what has been learned with phosphoproteomics.
l.368 – What is an “in-depth investigation”?
l.370 – “they identified 3,025 phosphopeptides” from which sample type?
l.370 – What do the authors mean by “altered phosphorylation levels”?
l.383 – “may have a positive effect” should be more specific.
l.387 – What do the authors mean by “in-depth analysis”?

Figure 2 is not cited in the text. As labelling strategies are not explained in the text, this figure could be very confusing and only provide little information to the reader. Additionally, the figures for metabolic labelling and chemical labelling are almost identical. For label-free quantification, it seems like the samples are being pooled, which should not be the case. Label-free quantification is often performed at the MS2 level as it is mostly performed on DIA data.

---

## Round 0.2 · Minor Revisions

Please revise your manuscript according to the reviewers remaining comments.

Yours,
Yoshi
Prof. Yoshinori Marunaka, M.D., Ph.D.

Reviewer 1 ·

Basic reporting

The revised version of this manuscript has enhanced the overall cohesion of the text, making it more suitable for the target audience. Additionally, the literature review has been updated to more accurately reflect the current state-of-the-art in phosphoproteomics techniques.

However, the figures are not very informative. Specifically, Figure 1 is referenced in the introduction, but its placement does not seem to align well with the text.

Moreover, the introduction to the state-of-the-art appears as a series of citations with little cohesion or clear explanation of their relevance. I suggest rephrasing lines 130 to 152 to improve coherence and better emphasize the significance of these works, rather than merely listing their titles.

Furthermore, minor phrasing issues are noted in point 4 ("Additional comments"). Some require correction as they convey a biased opinion on the ease of use of certain techniques compared to others, without a clear comparison of their respective advantages and disadvantages.

Experimental design

No comment.

Validity of the findings

The conclusions and discussions address current limitations but fail to suggest future directions for the evolution of phosphoproteomics to overcome these challenges.

Additional comments

Line 163: Proteases, as well as phosphatases, can be problematic during sample preparation.

Line 188: The authors summarize MS acquisition and use the term 'selective fragmentation.' Consider specifying 'of peptides' for clarity.

Line 207: The phrase 'easier set-up' expresses a biased opinion. DIA could be considered easier depending on the expertise of the MS handler.

Line 201 and Line 209: There is a repetition of sentences. Please revise to avoid redundancy.

Line 236: There is a biased opinion regarding the complexity of sample preparation for DIA analysis. It could be argued that DIA sample preparation is simpler since it does not rely on labeling for quantification, as in DDA-based techniques like TMT, SILAC, etc.

Line 245: IMS is mentioned here and again at the end of the paragraph. Please reorder for better readability.

Line 254: The authors state that PRM can provide more information than DIA. How did the authors reach this conclusion? Please correct or provide appropriate references; otherwise, this is a biased claim.

Reviewer 2 ·

Basic reporting

Altogether the reviewed manuscript has improved a lot in terms of clarity and bibliography. Some aspects still need to be improved.

• Introduction: ” Additionally, emerging mass spectrometry imaging techniques allow scientists to visually observe the distribution of phosphorylated proteins at the cellular and even tissue level, providing a powerful tool for studying the spatial heterogeneity of phosphorylated proteins13.” The reference used here is addressing spatial analysis in the case of renal disease, however it is is not including phosphoproteomics. Please remove the statement or find an appropriate citation supporting your claim. Additionally, MALDI is not an emerging technique.
• 2. Phosphoproteomics: Many relevant citations have been added to the paper, which is highly appreciated. However, it would be valuable to actually incorporate the citations within an introductory paragraph about phosphoproteomics rather than just pasting the citations one after the other, in the exact same order Reviewer 1 suggested them, which doesn’t make sense.
• 2.2. Mass spectrometry: The paragraph about MS has greatly improved. However, it should be streamlined as there are redundancies and some back and forth. One example is the mention of targeted approach, them IMS, then back to PRM and finally IMS is discussed.
• Figures: Even through the figures have improved, they still lack clarity for someone who is not an expert in proteomics or phosphoproteomics (target audience) to understand the workflows.
Figure 1:
C is not on the figure
The representation of the chromatogram is not really accurate.
What does the curve on the computer screen represents?
Figure 2:
The figure is still confusing and do not show show the differences between IMAC and MOAC as it was intended. What is the difference between the green pentagons and the light ellipses? I do not think that understanding the differences between IMAC and MOAC are relevant for the targeted audience and I would suggest the authors to find something else to plot. Alternatively, the differences should actually be highlighted.
Ti and Zr are very commonly used for IMAC-based phosphopeptide enrichment.
Additionally, the illustration of the MS is a Sciex QTRAP 4500 which is mainly used for the screening of small molecules and their metabolites, which is not the main type of mass spectrometer used for phosphoproteomics. Furthermore, this MS was released in 2012 and I would assume the authors could easily find a more recent model to illustrate their workflows.

Experimental design

No comment

Validity of the findings

No comment

Additional comments

I appreciate the effort in addressing the comments. However, I would like to note that the changes were not clearly specified in the rebuttal, and without the proper use of track changes (which doesn't consists in deleting the old text and pasting the new version), it has made it more difficult to follow the revisions. It would be helpful if the changes were either clearly indicated in the text with the appropriate use of track changes or explicitly detailed in your response to the reviewers.

---

## Round 0.3 · Major Revisions

Please respond to the reviewers' comments, and revise your manuscript.
Yours,
Prof. Yoshinori Marunaka, M.D., Ph.D.

Reviewer 1 ·

Basic reporting

No additional comments from previous revision.

Experimental design

No additional comments from previous revision.

Validity of the findings

The figure that accompanies the review has been updated in this revised paper, but now it includes several serious inconsistencies:
- Why the figure shows a nanodrop device to perform phosphosite quantification? This does not make sense. The review focuses on mass spectrometry as a quantative technique to quantify phosphosites. Therefore, showing a nanodrop for that purpose in the figure after the mass spectrometry step makes no technical sense.
- Similarly with the data analysis section of the figure: what does the "differential protein identification" panel refers to? That graph cannot relate to any kind of quantification derive from the type of mass spectrometry technology reported in the paper.

This figure cannot be published as it is right now since it is misleading and incorrect in the context of this review.

Additional comments

Even though the authors have address many of the previous concerns, the updated figure in this manuscript shows a significant failure to understand or grasp the technology they are describing. This figure needs to be corrected before the publication of this review.

Reviewer 2 ·

Basic reporting

The phosphoprotemics introduction paragraph has improved. I do understand that the authors here want to discuss the development in the field of phosphoproteomics leading to what is achievable now. However, this whole paragraph would be more impactful if the authors were to discuss those development in chronological order. For example, Leutert et al. have published their phosphopeptide enrichment automated protocol before the optimization carried out by Bortel et al, which is based on a similar protocol. Altogether I believe this paragraph would benefit from being streamlined again to enhance how an improvement led to the next and what impact it had on the field.

The outlook paragraph starting l.489 would also benefit in being streamlined. I do agree on the main points discussed which are (1) increasing the sensitivity in phosphoproteomics and (2) simplifying the workflows. However, I do not fully understand the logical connections within the paragraphs.

Figure 1 has been changed and figure 2 has been removed. However, I still believe that the newest version of figure 1 should be improved. This figure first contains very few information relative to a better understanding of the paper but also contains some errors. In the data acquisition part, I cannot see why or even how it would be possible to perform the phosphosite quantification using a nanodrop? The quantification is performed based on the MS data, which is a destructive technique. Additionally, the plots used for the statistical analysis are not representative of the type of analysis actually carried out on MS-based phosphoproteomics data or what the data can offer. Survival curves can be performed but not on the MS data, it has thus very few to do with phosphoproteomics workflows. Differential protein identification is carried out on the MS data and not based on absorbance measurements. Altogether the plots used for statistical analysis are based on the proteome level, however, this review is on phosphoproteomics and it should have been highlighted here. A more comprehensive legend than “phosphoproteomics workflow” would also be appreciated.

Experimental design

/

Validity of the findings

/

---

## Round 0.4 · accepted · Accept

Congratulations.

Yours,
Yoshi
Prof. Yoshinori Marunaka, M.D., Ph.D.

Reviewer 2 ·

Basic reporting

The comments from the previous round of revision have been addressed.
My only comments are about figure 1:
- There is a typo in "exyracellular" in the top right corner of the figure.
- The legend seem to be incomplete and stops with "Meanwhile, the Notch1 signalling". The authors need to complete the legend.

Experimental design

-

Validity of the findings

-

Additional comments

-